# Structured random receptive fields enable informative sensory encodings

**Biraj Pandey**[1], **Marius Pachitariu**[2], **Bingni W. Brunton**[3], **Kameron Decker Harris**[4]*

**1** Department of Applied Mathematics, University of Washington, Seattle, Washington, United States of America, **2** Janelia Research Campus, Howard Hughes Medical Institute, Ashburn, Virginia, United States of America, **3** Department of Biology, University of Washington, Seattle, Washington, United States of America, **4** Department of Computer Science, Western Washington University, Bellingham, Washington, United States of America

\* kameron.harris@wwu.edu

**Data Availability Statement:** All relevant data are within the manuscript and its Supporting information files.

**Funding:** B.P. was supported by a UW Applied Math Frederic Wan Endowed Fellowship, Terry

## Abstract

Brains must represent the outside world so that animals survive and thrive. In early sensory systems, neural populations have diverse receptive fields structured to detect important features in inputs, yet significant variability has been ignored in classical models of sensory neurons. We model neuronal receptive fields as random, variable samples from parameterized distributions and demonstrate this model in two sensory modalities using data from insect mechanosensors and mammalian primary visual cortex. Our approach leads to a significant theoretical connection between the foundational concepts of receptive fields and random features, a leading theory for understanding artificial neural networks. The modeled neurons perform a randomized wavelet transform on inputs, which removes high frequency noise and boosts the signal. Further, these random feature neurons enable learning from fewer training samples and with smaller networks in artificial tasks. This structured random model of receptive fields provides a unifying, mathematically tractable framework to understand sensory encodings across both spatial and temporal domains.

## Author summary

Evolution has ensured that animal brains are dedicated to extracting useful information from raw sensory stimuli while discarding everything else. Models of sensory neurons are a key part of our theories of how the brain represents the world. In this work, we model the tuning properties of sensory neurons in a way that incorporates randomness and builds a bridge to a leading mathematical theory for understanding how artificial neural networks learn. Our models capture important properties of large populations of real neurons presented with varying stimuli. Moreover, we give a precise mathematical formula for how sensory neurons in two distinct areas, one involving a gyroscopic organ in insects and the other visual processing center in mammals, transform their inputs. We also find that artificial models imbued with properties from real neurons learn more efficiently, with shorter training time and fewer examples, and our mathematical theory explains some of these findings. This work expands our understanding of sensory representation

Keegan Memorial ARCS Endowment Fellowship, and Natural Science Foundation Graduate Research Fellowship Program under Grant No. DGE-1762114. M.P. was supported by the Janelia Research Campus, Howard Hughes Medical Institute. B.W.B. was supported by grants FA9550-19-1-0386 & FA9550-18-1-0114 from the Air Force Office of Scientific Research. K.D.H. was supported by the Washington Research Foundation postdoctoral fellowship and Western Washington University. The funders had no role in study design, data collection and analysis, decision to publish, or preparation of the manuscript. https://amath.washington.edu/support-us https://www.arcsfoundation.org/national-homepage https://www.nsfgrfp.org https://www.janelia.org/lab/pachitariu-lab https://www.afrl.af.mil/AFOSR/ https://www.wrfseattle.org/grants/wrf-postdoctoral-fellowships/ https://cs.wwu.edu/harri267.

**Competing interests:** The authors have declared that no competing interests exist.

in large networks with benefits for both the neuroscience and machine learning communities.

## Introduction

It has long been argued that the brain uses a large population of neurons to represent the world [1–4]. In this view, sensory stimuli are encoded by the responses of the population, which are then used by downstream areas for diverse tasks, including learning, decision-making, and movement control. These sensory areas have different neurons responding to differing stimuli while also providing a measure of redundancy. However, we still lack a clear understanding of what response properties are well-suited for different sensory modalities.

One way to approach sensory encoding is by understanding how a neuron would respond to arbitrary stimuli. Experimentally, we typically present many stimuli to the animal, measure the responses of sensory neurons, then attempt to estimate some kind of model for how the neurons respond to an arbitrary stimulus. A common assumption is that the neuron computes a linear filter of the stimulus, which then drives spiking through a nonlinear spike-generating mechanism. Mathematically, this assumption can be summarized as the number of measured spikes for a stimulus $x$ being equal to $\sigma(w^T x)$ for a weight vector $w$ and nonlinearity $\sigma$. Here, the weights $w$ define the filtering properties of the neuron, also known as its *receptive field* [5]. This model is known as a *linear-nonlinear* (LN) model [6], and it is also the most common form of artificial neuron in artificial neural networks (ANNs). LN models have been used extensively to describe the firing of diverse neurons in various sensory modalities of vertebrates and invertebrates. In the mammalian visual system, LN models have been used to characterize retinal ganglion cells [7], lateral geniculate neurons [8], and simple cells in primary visual cortex (V1) [9]. They have also been used to characterize auditory sensory neurons in the avian midbrain [10] and somatosensory neurons in the cortex [11]. In insects, they have been used to understand the response properties of visual interneurons [12], mechanosensory neurons involved in proprioception [13, 14], and auditory neurons during courtship behavior [15].

Given the stimulus presented and neural response data, one can then estimate the receptive fields of a population of neurons. Simple visual receptive fields have classically been understood as similar to wavelets with particular spatial frequency and angular selectivity [9]. In mechanosensory areas, receptive fields are selective to temporal frequency over a short time window [13]. Commonly, parametric modeling (Gabor wavelets [4]) or smoothing (regularization, etc. [16]) are used to produce "clean" receptive fields. Yet, the data alone show noisy receptive fields that are perhaps best modeled using a random distribution [17]. As we will show, modeling receptive fields as random samples produces realistic receptive fields that reflect both the structure and noisiness seen in experimental data. More importantly, this perspective creates significant theoretical connections between foundational ideas from neuroscience and artificial intelligence. This connection helps us understand why receptive fields have the structures that they do and how this structure relates to the kinds of stimuli that are relevant to the animal.

Modeling the filtering properties of a population of LN neurons as samples from a random distribution leads to the study of networks with random weights [18–20]. In machine learning (ML), such networks are known as *random feature networks* (RFNs) [21–24]. The study of RFNs has rapidly gained popularity in recent years, in large part because it offers a theoretically tractable way to study the learning properties of ANNs where the weights are tuned using data

[25–27]. When the RFN contains many neurons, it can approximate functions that live in a well-understood function space. This function space is called a *reproducing kernel Hilbert space* (RKHS), and it depends on the network details, in particular the weight i.e. receptive field distribution [28–30]. Learning can then be framed as approximating functions in this space from limited data.

Several recent works highlight the RFN theory's usefulness for understanding learning in neural systems. Bordelon, Canatar, and Pehlevan, in a series of papers, have shown that neural codes allow learning from few examples when spectral properties of their second-order statistics aligns with the spectral properties of the task [31–33]. When applied to V1, they found that the neural code is aligned with tasks that depend on low spatial frequency components. Harris constructed an RFN model of sparse networks found in associative centers like the cerebellum and insect mushroom body and showed that these areas may behave like additive kernels [34], an architecture also considered by Hashemi et al. [35]. These classes of kernels are beneficial for learning in high dimensions because they can learn from fewer examples and remain resilient to input noise or adversarial perturbation. Xie et al. investigated the relationship between the fraction of active neurons in a model of the cerebellum—controlled by neuron thresholds—and generalization performance for learning movement trajectories [36]. In the vast majority of network studies with random weights, these weights $w$ are drawn from a Gaussian distribution with independent entries. This sampling is equivalent to a fully *unstructured* receptive field, which looks like white noise.

Closely related to our work, a previous study of ANNs showed that directly learning structured receptive fields could improve image classification in deep networks [37]. Their receptive fields were parametrized as a sum of Gaussian derivatives up to fourth order. This led to better performance against rival architectures in low data regimes.

In this paper, we study the effect of having *structured yet random* receptive fields and how they lead to informative sensory encodings. Specifically, we consider receptive fields generated by a Gaussian process (GP), which can be thought of as drawing the weights $w$ from a Gaussian distribution with a particular covariance matrix. We show that networks with such random weights project the input to a new basis and filter out particular components. This theory introduces realistic structure of receptive fields into random feature models which are crucial to our current understanding of artificial networks. Next, we show that receptive field datasets from two disparate sensory systems, mechanosensory neurons on insect wings and V1 cortical neurons from mice and monkeys, are well-modeled by GPs with covariance functions that have wavelet eigenbases. Given the success of modeling these data with the GP, we apply these weight distributions in RFNs that are used in synthetic learning tasks. We find that these structured weights improve learning by reducing the number of training examples and the size of the network needed to learn the task. Thus, structured random weights offer a realistic generative model of the receptive fields in multiple sensory areas, which we understand as performing a random change of basis. This change of basis enables the network to represent the most important properties of the stimulus, which we demonstrate to be useful for learning.

## Results

We construct a generative model for the receptive fields of sensory neurons and use it for the weights of an ANN. We refer to such a network as a *structured* random feature network. We first review the basics of random feature networks, the details and rationale behind our generative model, and the process by which we generate hidden weights. Our main theory result is that networks with such weights transform the inputs into a new basis and filter out particular components, thus bridging sensory neuroscience and the theory of neural networks. Next, we

show that neurons in two receptive field datasets—insect mechanosensory neurons and mammalian V1 cortical neurons—are well-described by our generative model. There is a close resemblance between the the second-order statistics, sampled receptive fields, and their principal components for both data and model. Finally, we show the performance of structured random feature networks on several synthetic learning tasks. The hidden weights from our generative model allows the network to learn from fewer training examples and smaller network sizes.

## Theoretical analysis

We consider receptive fields generated by GPs in order to connect this foundational concept from sensory neuroscience to the theory of random features in artificial neural networks. GPs can be thought of as samples from a Gaussian distribution with a particular covariance matrix, and we initialize the hidden weights of RFNs using these GPs. We show that using a GP causes the network to project the input into a new basis and filter out particular components. The basis itself is determined by the covariance matrix of the Gaussian, and is useful for removing irrelevant and noisy components from the input. We use these results to study the space of functions that RFNs containing many neurons can learn by connecting our construction to the theory of kernel methods.

**Random feature networks.** We start by introducing the main learning algorithm and the neuronal model of our work, the RFN. Consider a two-layer, feedforward ANN. Traditionally, all the weights are initialized randomly and learned through backpropagation by minimizing some loss objective. In sharp contrast, RFNs have their hidden layer weights sampled randomly from some distribution and fixed. Each hidden unit computes a random feature of the input, and only the output layer weights are trained (Fig 1). Note that the weights are randomly drawn but the neuron's response is a deterministic function of the input given the weights.

Mathematically, we have the hidden layer activations and output given by

$$\boldsymbol{h} = \sigma(\boldsymbol{W}\boldsymbol{x}), \qquad \hat{y} = \boldsymbol{\beta}^T \boldsymbol{h} + \beta_0, \tag{1}$$

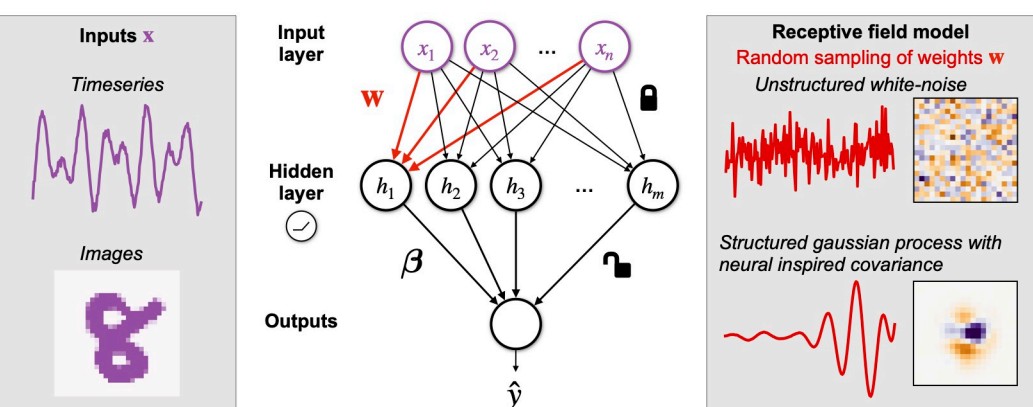

**Fig 1. Random feature networks with structured weights.** We study random feature networks as models for learning in sensory regions. In these networks, each neuron's weight $\boldsymbol{w}$ is fixed as a random sample from some specified distribution. Only the readout weights $\boldsymbol{\beta}$ are trained. In particular, we specify distributions to be Gaussian Processes (GPs) whose covariances are inspired by biological neurons; thus, each realization of the GP resembles a biological receptive field. We build GP models of two sensory areas that specialize in processing timeseries and image inputs. We initialize $\boldsymbol{w}$ from these *structured* GPs and compare them against initialization from *unstructured* white-noise distribution.

where $\boldsymbol{x} \in \mathbb{R}^d$ is the stimulus, $\boldsymbol{h} = [h_1, h_2, \ldots, h_m]^T \in \mathbb{R}^m$ are the hidden neuron responses, and $\hat{y} \in \mathbb{R}$ is the predicted output. We use a rectified linear (ReLU) nonlinearity, $\sigma(x) = \max(0, x)$ applied entrywise in Eq (1). The hidden layer weights $\boldsymbol{W} = [\boldsymbol{w}_1, \boldsymbol{w}_2, \ldots, \boldsymbol{w}_m]^T \in \mathbb{R}^{m \times d}$ are drawn randomly and fixed. Only the readout weights $\beta_0$ and $\boldsymbol{\beta}$ are trained in RFNs.

In our RFN experiments, we train the readout weights $\boldsymbol{\beta} \in \mathbb{R}^m$ and offset $\beta_0 \in \mathbb{R}$ using a support vector machine (SVM) classifier with squared hinge loss and $\ell^2$ penalty with regularization strength of tuned in the range $[10^{-3}, 10^3]$ by 5-fold cross-validation. Our RFNs do not include a threshold for the hidden neurons, although this could help in certain contexts [36].

In the vast majority of studies with RFNs, each neuron's weights $\boldsymbol{w} \in \mathbb{R}^d$ are initialized i.i.d. from a spherical Gaussian distribution $\boldsymbol{w} \sim \mathcal{N}(0, \boldsymbol{I}_d)$. We will call networks built this way *classical unstructured* RFNs (Fig 1). We propose a variation where hidden weights are initialized $\boldsymbol{w} \sim \mathcal{N}(0, \boldsymbol{C})$, where $\boldsymbol{C} \in \mathbb{R}^{d \times d}$ is a positive semidefinite covariance matrix. We call such networks *structured* RFNs (Fig 1), to mean that the weights are random with a specified covariance. To compare unstructured and structured weights on equal footing, we normalize the covariance matrices so that $\text{Tr}(\boldsymbol{C}) = \text{Tr}(\boldsymbol{I}_d) = d$, which ensures that the mean square amplitude of the weights $\mathbb{E}[\|\boldsymbol{w}\|^2] = d$.

**Receptive fields modeled by linear weights.**   Sensory neurons respond preferentially to specific features of their inputs. This stimulus selectivity is often summarized as a neuron's receptive field, which describes how features of how the sensory space elicits responses when stimulated [5]. Mathematically, receptive fields are modeled as a linear filter in the stimulus space. Linear filters are also an integral component of the widely used LN model of sensory processing [6]. According to this model, the firing rate of a neuron is a nonlinear function applied to the projection of the stimulus onto the low-dimensional subspace of the linear filter.

A linear filter model of receptive fields can explain responses of individual neurons to diverse stimuli. It has been used to describe disparate sensory systems like visual, auditory, and somatosensory systems of diverse species including birds, mammals, and insects [7, 10–12, 15]. If the stimuli are uncorrelated, the filters can be estimated by computing the spike triggered average (STA), the average stimulus that elicited a spike for the neuron. When the stimuli are correlated, the STA filter is whitened by the inverse of the stimulus covariance matrix [38]. Often these STAs are denoised by fitting a parametric function to the STA [6], such as Gabor wavelets for simple cells in V1 [9].

We model the receptive field of a neuron $i$ as its weight vector $\boldsymbol{w}_i$ and its nonlinear function as $\sigma$. Instead of fitting a parametric function, we construct covariance functions so that each realization of the resulting Gaussian process resembles a biological receptive field (Fig 1).

**Structured weights project and filter input into the covariance eigenbasis.**   We generate network weights from Gaussian processes (GP) whose covariance functions are inspired by the receptive fields of sensory neurons in the brain. By definition, a GP is a stochastic process where finite observations follow a Gaussian distribution [39]. We find that networks with such weights project inputs into a new basis and filter out irrelevant components. We will see that this adds an inductive bias to classical RFNs for tasks with naturalistic inputs and improves learning.

We view our weight vector $\boldsymbol{w}$ as the finite-dimensional discretization of a continuous function $w(t)$ which is a sample from a GP. The continuous function has domain $T$, a compact subset of $\mathbb{R}^D$, and we assume that $T$ is discretized using a grid of $d$ equally spaced points $\{t_1, \ldots, t_d\} \subset T$, so that $w_i = w(t_i)$. Let the input be a real-valued function $x(t)$ over the same domain $T$, which could represent a finite timeseries ($D = 1$), an image of luminance on the retina ($D = 2$), or more complicated spatiotemporal sets like a movie ($D = 3$). In the continuous setting, the $d$-dimensional $\ell^2$ inner product $\boldsymbol{w}^T \boldsymbol{x} = \sum_{i=1}^{d} w_i x_i$ gets replaced by the $L^2(T)$ inner product $\langle w, x \rangle = \int_{t \in T} w(t) x(t) \mathrm{d}t$.

Every GP is fully specified by its mean and covariance function $C(t, t')$. We will always assume that the mean is zero and study different covariance functions. By the Kosambi-Karhunen–Loève theorem [40], each realization of a zero-mean GP has a random series representation

$$w(t) = \sum_{i=1}^{\infty} z_i \lambda_i \phi_i(t),\tag{2}$$

in terms of standard Gaussian random variables $z_i \sim \mathcal{N}(0, 1)$, functions $\phi_i(t)$, and weights $\lambda_i \geq 0$. The pairs $(\lambda_i^2, \phi_i)$ are eigenvalue, eigenfunction pairs of the covariance operator $\mathcal{C} : L^2(T) \to L^2(T)$,

$$(\mathcal{C}f)(t) = \int_{t \in T} C(t, t')f(t')dt',$$

which is the continuous analog of the covariance matrix $\boldsymbol{C}$. If $C(t, t')$ is positive definite, as opposed to just semidefinite, all $\lambda_i^2 > 0$ and these eigenfunctions $\phi_i$ form a complete basis for $L^2(T)$. Using Eq (2), the inner product between a stimulus and a neuron's weights is

$$\langle w, x \rangle = \left\langle \sum_{i=1}^{\infty} z_i \lambda_i \phi_i, x \right\rangle = \sum_{i=1}^{\infty} z_i \lambda_i \langle \phi_i, x \rangle = \sum_{i=1}^{\infty} z_i \tilde{x}_i, \quad \text{where } \tilde{x}_i = \lambda_i \langle \phi_i, x \rangle.\tag{3}$$

Eq (3) shows that the structured weights compute a *projection* of the input $x$ onto each eigenfunction $\langle \phi_i, x \rangle$ and reweight or *filter* by the eigenvalue $\lambda_i$ before taking the $\ell^2$ inner product with the random Gaussian weights $z_i$.

It is illuminating to see what these continuous equations look like in the $d$-dimensional discrete setting. Samples from the finite-dimensional GP are used as the hidden weights in RFNs, $\boldsymbol{w} \sim \mathcal{N}(0, \boldsymbol{C})$. First, the GP series representation Eq (2) becomes $\boldsymbol{w} = \boldsymbol{\Phi}\boldsymbol{\Lambda}\boldsymbol{z}$, where $\boldsymbol{\Lambda}$ and $\boldsymbol{\Phi}$ are matrices of eigenvalues and eigenvectors, and $\boldsymbol{z} \sim \mathcal{N}(0, \boldsymbol{I}_d)$ is a Gaussian random vector. By the definition of the covariance matrix, $\boldsymbol{C} = \mathbb{E}[\boldsymbol{w}\boldsymbol{w}^T]$, which is equal to $\boldsymbol{\Phi}\boldsymbol{\Lambda}^2\boldsymbol{\Phi}^T$ after a few steps of linear algebra. Finally, Eq (3) is analogous to $\boldsymbol{w}^T\boldsymbol{x} = \boldsymbol{z}^T\boldsymbol{\Lambda}\boldsymbol{\Phi}^T\boldsymbol{x}$. Since $\boldsymbol{\Phi}$ is an orthogonal matrix, $\boldsymbol{\Phi}^T\boldsymbol{x}$ is equivalent to a change of basis, and the diagonal matrix $\boldsymbol{\Lambda}$ shrinks or expands certain directions to perform filtering. This can be summarized in the following theorem:

**Theorem 1 (Basis change formula)** *Assume $\boldsymbol{w} \sim \mathcal{N}(0, \boldsymbol{C})$ with $\boldsymbol{C} = \boldsymbol{\Phi}\boldsymbol{\Lambda}^2\boldsymbol{\Phi}^T$ its eigenvalue decomposition. For $\boldsymbol{x} \in \mathbb{R}^d$, define*

$$\tilde{\boldsymbol{x}} := \boldsymbol{\Lambda}\boldsymbol{\Phi}^T\boldsymbol{x}.\tag{4}$$

*Then $\boldsymbol{w}^T\boldsymbol{x} = \boldsymbol{z}^T\tilde{\boldsymbol{x}}$ for $\boldsymbol{z} \sim \mathcal{N}(0, \boldsymbol{I}_d)$.*

Theorem 1 says that projecting an input onto a structured weight vector is the same as first filtering that input in the GP eigenbasis and doing a random projection onto a spherical random Gaussian. The form of the GP eigenbasis is determined by the choice of the covariance function. If the covariance function is compatible with the input structure, the hidden weights filter out any irrelevant features or noise in the stimuli while amplifying the descriptive features. This inductive bias facilitates inference on the stimuli by any downstream predictor. Because the spherical Gaussian distribution is the canonical choice for unstructured RFNs, there is a simple way to evaluate the effective kernel of structured RFNs as $k_{\text{struct}}(\boldsymbol{x}, \boldsymbol{x}') = k_{\text{unstruct}}(\tilde{\boldsymbol{x}}, \tilde{\boldsymbol{x}}')$ (see S1 Appendix).

Our expression for the structured kernel provides a concrete connection to the kernel theory of learning using nonlinear neural networks. For readers interested in such kernel theories, a full example and simulation results of how these work is given in S1 Appendix. There we

show that there can be an exponential reduction in the number of samples needed to learn frequency detection using a structured versus unstructured basis (Fig A in S1 Appendix).

## Examples of random yet structured receptive fields

Our goal is to model the weights of artificial neurons in a way that is inspired by biological neurons' receptive fields. Structured RFNs sample hidden weights from GPs with structured covariance, so we construct covariance functions that make the generated weights resemble neuronal receptive fields. We start with a toy example of a stationary GP with well-understood Fourier eigenbasis and show how the receptive fields generated from this GP are selective to frequencies in timeseries signals. Then, we construct locally stationary covariance models of the of insect mechanosensory and V1 neuron receptive fields. These models are shown to be a good match for experimental data.

**Warm-up: Frequency selectivity from stationary covariance.**   To illustrate some results from our theoretical analysis, we start with a toy example of temporal receptive fields that are selective to particular frequencies. This example may be familiar to readers comfortable with Fourier series and basic signal processing. Let the input be a finite continuous timeseries $x(t)$ over the interval $T = [0, L]$. We use the covariance function

$$C(t, t') \quad = \overbrace{\sum_{k=0}^{\infty} \lambda_k^2 \cos(\omega_k(t - t'))}^{\text{stationary process}}, \tag{5}$$

where $\omega_k = 2\pi k/L$ is the $k$th natural frequency and $\lambda_k^2$ are the weight coefficients. The covariance function Eq (5) is *stationary*, which means that it only depends on the difference between the timepoints $t - t'$. Applying the compound angle formula, we get

$$C(t, t') = \sum_{k=0}^{\infty} \lambda_k^2 (\cos(\omega_k t)\cos(\omega_k t') + \sin(\omega_k t)\sin(\omega_k t')). \tag{6}$$

Since the sinusoidal functions $\cos(\omega_k t)$ and $\sin(\omega_k t)$ form an orthonormal basis for $L^2(T)$, Eq (6) is the eigendecomposition of the covariance, where the eigenfunctions are sines and cosines with eigenvalues $\lambda_k^2$. From Eq (2), we know that structured weights with this covariance form a random series:

$$w(t) = \sum_{k=0}^{\infty} \lambda_k (z_k \cos(\omega_k t) + z_k' \sin(\omega_k t)), \tag{7}$$

where each $z_k, z_k' \sim \mathcal{N}(0, 1)$. Thus, the receptive fields are made up of sinusoids weighted by $\lambda_k$ and the Gaussian variables $z_k, z_k'$.

Suppose we want receptive fields that only retain specific frequency information of the signal and filter out the rest. Take $\lambda_k = 0$ for any $k$ where $\omega_k < f_{\text{lo}}$ or $\omega_k > f_{\text{hi}}$. We call this a *bandlimited* spectrum with passband $[f_{\text{lo}}, f_{\text{hi}}]$ and bandwidth $f_{\text{hi}} - f_{\text{lo}}$. As the bandwidth increases, the receptive fields become less smooth since they are made up of a wider range of frequencies. If the $\lambda_k$ are all nonzero but decay at a certain rate, this rate controls the smoothness of the resulting GP [41].

When these receptive fields act on input signals $x(t)$, they implicitly transform the inputs into the Fourier basis and filter frequencies based on the magnitude of $\lambda_k$. In a bandlimited setting, any frequencies outside the passband are filtered out, which makes the receptive fields selective to a particular range of frequencies and ignore others. On the other hand, classical

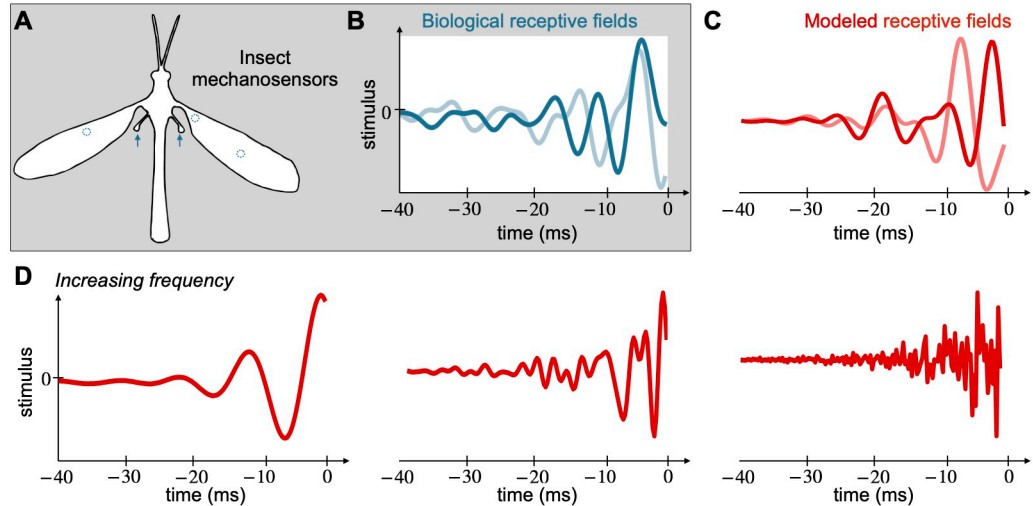

**Fig 2. Random receptive field model of insect mechanosensors.** (A) Diagram of the the cranefly, *Tipula hespera*. Locations of the mechanosensors, campaniform sensilla, are marked in blue on the wings and halteres. (B) Two receptive fields of campaniform sensilla are shown in blue. They are smooth, oscillatory, and decay over time. We model them as random samples from distributions parameterized by frequency and decay parameters. Data are from the hawkmoth [14]; cranefly sensilla have similar responses [13]. (C) Two random samples from the model distribution are shown in red. (D) The smoothness of the receptive fields is controlled by the frequency parameter. The decay parameter controls the rate of decay from the origin (not shown).

random features weight all frequencies equally, even though in natural settings high frequency signals are the most corrupted by noise.

**Insect mechanosensors.**   We next consider a particular biological sensor that is sensitive to the time-history of forces. Campaniform sensilla (CS) are dome-shaped mechanoreceptors that detect local stress and strain on the insect exoskeleton [42]. They are embedded in the cuticle and deformation of the cuticle through bending or torsion induces depolarizing currents in the CS by opening mechanosensitive ion channels. The CS encode proprioceptive information useful for body state estimation and movement control during diverse tasks like walking, kicking, jumping, and flying [42].

We will model the receptive fields of CS that are believed to be critical for flight control, namely the ones found at the base of the halteres [43] and on the wings [14] (Fig 2A). Halteres and wings flap rhythmically during flight, and rotations of the insect's body induce torsional forces that can be felt on these active sensory structures. The CS detect these small strain forces, thereby encoding angular velocity of the insect body [43]. Experimental results show haltere and wing CS are selective to a broad range of oscillatory frequencies [14, 44], with STAs that are smooth, oscillatory, selective to frequency, and decay over time [13] (Fig 2B).

We model these temporal receptive fields with a locally stationary GP [45] with bandlimited spectrum. Examples of receptive fields generated from this GP are shown in Fig 2C. The inputs to the CS are modeled as a finite continuous timeseries $x(t)$ over the finite interval $T = [0, L]$. The neuron weights are generated from a covariance function

$$C(t, t') = \overbrace{\exp\left(-\frac{(t + t')}{\gamma}\right)}^{\text{localized}} \overbrace{\sum_{k=0}^{\infty} \lambda_k^2 \cos(\omega_k(t - t'))}^{\text{stationary process}}, \qquad \lambda_k = \overbrace{\begin{cases} 1 & f_{\text{lo}} \leq \omega_k \leq f_{\text{hi}} \\ 0 & \text{otherwise} \end{cases}}^{\text{bandlimited, flat-power spectrum}}, \quad (8)$$

where $\omega_k = 2\pi k/L$ is the $k$th natural frequency. As in the warmup, the frequency selectivity of

their weights is accounted for by the parameters $f_{lo}$ and $f_{hi}$. As the bandwidth $f_{hi} - f_{lo}$ increases, the receptive fields are built out of a wider selection of frequencies. This makes the receptive fields less smooth (Fig 2D). Each field is localized to near $t = 0$, and its decay with $t$ is determined by the parameter $\gamma$. As $\gamma$ increases, the receptive field is selective to larger time windows.

The eigenbasis of the covariance function Eq (8) is similar to a Fourier eigenbasis modulated by a decaying exponential. The eigenbasis is an orthonormal basis for the span of $\lambda_k e^{-t/\gamma} \cos(\omega_k t)$ and $\lambda_k e^{-t/\gamma} \sin(\omega_k t)$, which are a non-orthogonal set of functions in $L^2(T)$. The hidden weights transform timeseries inputs into this eigenbasis and discard components outside the passband frequencies $[f_{lo}, f_{hi}]$.

We fit the covariance model to receptive field data from 95 CS neurons from wings of the hawkmoth *Manduca sexta* (data from [14]). Briefly, CS receptive fields were estimated as the spike-triggered average (STA) of experimental mechanical stimuli of the wings, where the stimuli were generated as bandpassed white noise (2–300 Hz).

To characterize the receptive fields of this population of CS neurons, we compute the data covariance matrix $C_{data}$ by taking the inner product between the receptive fields. We normalize the trace to be the dimension of each receptive field (number of samples), which in this case is 40 kHz × 40 ms = 1600 samples. This normalization sets the overall scale of the covariance matrix. The data covariance matrix shows a tridiagonal structure (Fig 3A). The main diagonal is positive while the off diagonals are negative. All diagonals decay away from the top left of the matrix.

To fit the covariance model to the data, we optimize the parameters $f_{lo}$, $f_{hi}$, and $\gamma$, finding $f_{lo} = 75$ Hz, $f_{hi} = 200$ Hz, and $\gamma = 12.17$ ms best fit the sensilla data. We do so by minimizing

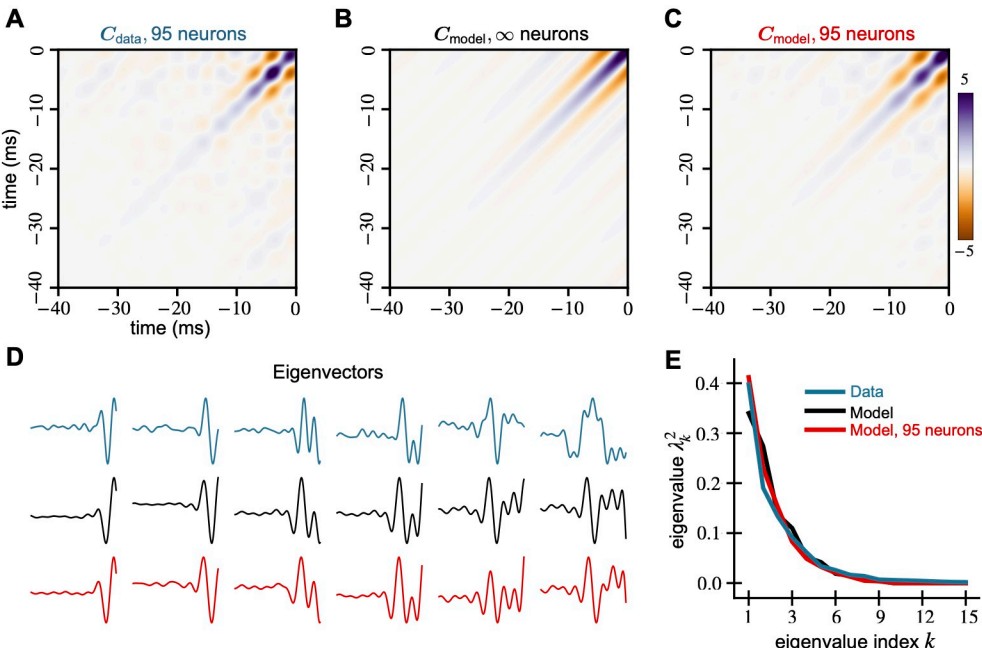

**Fig 3. Spectral properties of mechanosensory RFs and our model are similar.** We compare the covariance matrices generated from (A) receptive fields of 95 mechanosensors from [14], (B) the model Eq (8), and (C) 95 random samples from the same model. All covariance matrices show a tri-diagonal structure that decays away from the origin. (D) The first five principal components of all three covariance matrices are similar and explain 90% of the variance in the RF data. (E) The leading eigenvalue spectra of the data and models show similar behavior.

the Frobenius norm of the difference between $C_{\text{data}}$ and the model (see S1 Appendix). The resulting model covariance matrix (Fig 3B) matches the data covariance matrix (Fig 3A) remarkably well qualitatively. The normalized Frobenius norm of the difference between $C_{\text{data}}$ and the model is 0.4386. Examples of biological receptive fields and random samples from this fitted covariance model are shown in Fig B in S1 Appendix. To simulate the effect of a finite number of neurons, we generate 95 weight vectors (equal to the number of neurons recorded) and recompute the model covariance matrix (Fig 3C). We call this the finite neuron model covariance matrix $C_{\text{finite}}$, and it shows the bump and blob-like structures evident in $C_{\text{data}}$ but not in $C_{\text{model}}$. This result suggests that these bumpy structures can be attributed to having a small number of recorded neurons. We hypothesize that these effects would disappear with a larger dataset and $C_{\text{data}}$ would more closely resemble $C_{\text{model}}$.

For comparison, we also calculate the Frobenius difference for null models, the unstructured covariance model and the Fourier model (5). For the unstructured model, the Frobenius norm difference is 0.9986 while that of the Fourier model is 0.9123. The sensilla covariance model has a much lower difference (0.4386) compared to the null models, fitting the data more accurately. We show the covariance matrices and sampled receptive fields from the null models in Fig C to E in S1 Appendix.

Comparing the eigenvectors and eigenvalues of the data and model covariance matrices, we find that the spectral properties of both $C_{\text{model}}$ and $C_{\text{finite}}$ are similar to that of $C_{\text{data}}$. The eigenvalue curves of the models match that of the data quite well (Fig 3E); these curves are directly comparable because each covariance is normalized by its trace, which makes the sum of the eigenvalues unity. Further, all of the data and the model covariance matrices are low-dimensional. The first 10 data eigenvectors explain 97% of the variance, and the top 5 explain 90%. The top 5 eigenvectors of the model and its finite sample match that of the data quite well (Fig 3D).

**Primary visual cortex.**   We now turn to visually driven neurons from the mammalian primary cortex. Primary visual cortex (V1) is the earliest cortical area for processing visual information (Fig 4A). The neurons in V1 can detect small changes in visual features like orientations, spatial frequencies, contrast, and size.

Here, we model the receptive fields of simple cells in V1, which have clear excitatory and inhibitory regions such that light shone on the excitatory regions increase the cell's response and vice-versa (Fig 4B). The shape of the regions determines the orientation selectivity, while their widths determine the frequency selectivity. The receptive fields are centered to a location in the visual field and decay away from it. They integrate visual stimuli within a small region of this center [46]. Gabor functions are widely used as a mathematical model of the receptive fields of simple cells [9].

We model these receptive fields using another locally stationary GP [45] and show examples of generated receptive fields in Fig 4C. Consider the inputs to the cortical cells to be a continuous two-dimensional image $x(\boldsymbol{t})$, where the domain $T = [0, L] \times [0, L']$ and $x : T \to \mathbb{R}$. Since the image is real-valued, $x(\boldsymbol{t})$ is the grayscale contrast or single color channel pixel values. The neuron weights are then generated from a covariance function of the following form:

$$C(\boldsymbol{t}, \boldsymbol{t}') = \overbrace{\exp\left(-\frac{\|\boldsymbol{t} - \boldsymbol{t}'\|^2}{2f^2}\right)}^{\text{smooth receptive fields}} \cdot \overbrace{\exp\left(-\frac{\|\boldsymbol{t} - \boldsymbol{c}\|^2 + \|\boldsymbol{t}' - \boldsymbol{c}\|^2}{2s^2}\right)}^{\text{localized to a center } c}. \qquad (9)$$

The receptive field center is defined by $\boldsymbol{c}$, and the size of the receptive field is determined by the parameter $s$. As $s$ increases, the receptive field extends farther from the center $\boldsymbol{c}$ (Fig 4D).

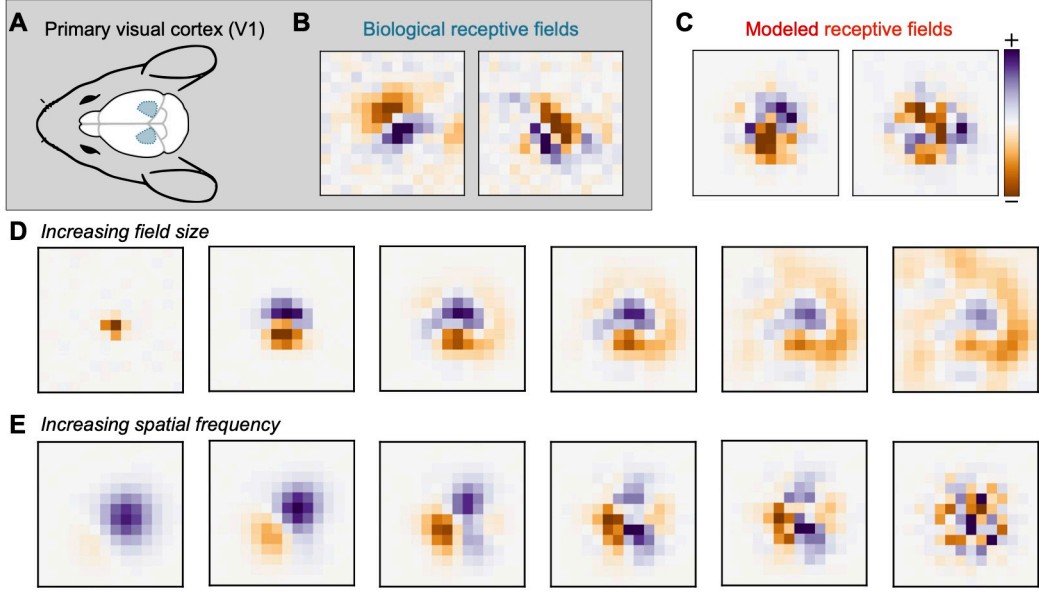

**Fig 4. Random receptive field model of Primary Visual Cortex (V1).** (A) Diagram of the mouse brain with V1 shown in blue. (B) Receptive fields of two mouse V1 neurons calculated from their response to white noise stimuli. The fields are localized to a region in a visual field and show "on" and "off" regions. (C) Random samples from the model Eq (9) distribution. (D) Increasing the receptive field size parameter in our model leads to larger fields. (E) Increasing the model spatial frequency parameter leads to more variable fields.

Spatial frequency selectivity is accounted for by the bandwidth parameter *f*. As *f* decreases, the spatial frequency of the receptive field goes up, making the weights less smooth (Fig 4E).

The eigendecomposition of the covariance function Eq (9) leads to an orthonormal basis of single scale *Hermite wavelets* [47, 48]. When *c* = 0, the wavelet eigenfunctions are Hermite polynomials modulated by a decaying Gaussian:

$$\phi_{\boldsymbol{k}}(\boldsymbol{t}) \propto \prod_{i=1}^{D} e^{-c_1 t_i^2} H_{k_i}(c_2 t_i) \quad \text{and} \quad \lambda_{\boldsymbol{k}}^2 \propto \prod_{i=1}^{D} c_3^{k_i}, \tag{10}$$

where $H_k$ is the *k*th (physicist's) Hermite polynomial; eigenfunctions for nonzero centers *c* are just shifted versions of Eq (10). The detailed derivation and values of the constants $c_1, c_2, c_3$ and normalization are in S1 Appendix.

We use Eq (9) to model receptive field data from 8,358 V1 neurons recorded with calcium imaging from transgenic mice expressing GCaMP6s; the mice were headfixed and running on an air-floating ball. We presented 24,357 unique white noise images of 14 × 36 pixels using the Psychtoolbox [49], where the pixels were white or black with equal probability. Images were upsampled to the resolution of the screens via bilinear interpolation. The stimulus was corrected for eye-movements online using custom code. The responses of 45,026 cells were collected using a two-photon mesoscope [50] and preprocessed using Suite2p [51]. Receptive fields were calculated from the white noise images and the deconvolved calcium responses of the cells using the STA. For the covariance analysis, we picked cells above the signal-to-noise (SNR) threshold of 0.4; this gave us 8,358 cells. The SNR was calculated from a smaller set of 2,435 images that were presented twice using the method from [4]. As a preprocessing step, we moved the center of mass of every receptive field to the center of the visual field.

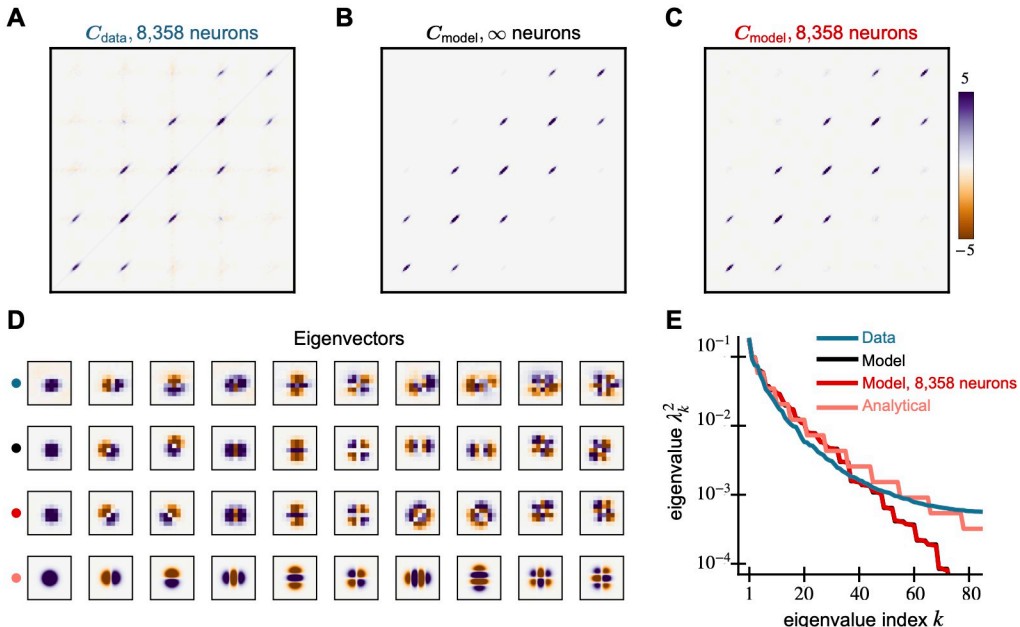

**Fig 5. Spectral properties of V1 RFs and our model are similar.** We compare the covariance matrices generated from the (A) receptive fields of 8,358 mouse V1 neurons, (B) the GP model Eq (9), and (C) 8,358 random samples from the model. These resemble a tri-diagonal matrix whose diagonals are non-zero at equally-spaced segments. (D) The leading 10 eigenvectors of the data and model covariance matrices show similar structure and explain 68% of the variance in the data. Analytical Hermite wavelet eigenfunctions are in the last row and differ from the model due to discretization (both cases) and finite sampling (8,358 neurons only). (E) The eigenspectrum of the model matches well with the data. The staircase pattern in the model comes from repeated eigenvalues at each frequency. The model curve with infinite neurons (black) is obscured by the model curve with 8,358 neurons (red).

We compute the data covariance matrix $C_{\text{data}}$ by taking the inner product between the receptive fields. We normalize the trace to be the dimension of each receptive field, which in this case is $(14 \times 36)$ pixels = 504 pixels. The data covariance matrix resembles a tridiagonal matrix. However, the diagonals are non-zero only at equally spaced segments. Additionally, their values decay away from the center of the matrix. We show $C_{\text{data}}$ zoomed in at the non-zero region around the center of the matrix (Fig 5A); this corresponds to the $180 \times 180$ pixel region around the center of the full $504 \times 504$ pixel matrix. The full covariance matrix is shown in Fig F in S1 Appendix.

In the covariance model, the number of off-diagonals, the center, the rate of their decay away from the center are determined by the parameters $f$, $s$ and $c$ respectively. The covariance between pixels decays as a function of their distance from $c$. This leads to the the equally-spaced non-zero segments. On the other hand, the covariance also decays as a function of the distance between pixels. This brings the the diagonal structure to the model. When the frequency parameter $f$ increases, the number of off-diagonals increases. Pixels in the generated weights become more correlated and the weights become spatially smoother. When the size parameter $s$ increases, the diagonals decay slower from the center $c$, increasing correlations with the center pixel and leading the significant weights to occupy more of the visual field.

We again optimize the parameters to fit the data, finding $s = 1.87$ and $f = 0.70$ pixels, by minimizing the Frobenius norm of the difference between $C_{\text{data}}$ and the model. We do not need to optimize over the center parameter $c$, since we preprocess the data so that all receptive fields are centered at $c = (7, 18)$, the center of the $14 \times 36$ grid. The resulting model covariance

matrix (Fig 5B) and the data covariance matrix (Fig 5A) match remarkably well qualitatively. The normalized Frobenius norm of the difference between $C_{data}$ and the model is 0.2993. Examples of biological receptive fields and random samples from the fitted covariance model are shown in Fig G in S1 Appendix. To simulate the effect of a finite number of neurons, we generate 8,358 weights, equal to the number of neurons in our data, to compute $C_{finite}$ shown in Fig 5C. This finite matrix $C_{finite}$ looks even more like $C_{data}$, and it shows that some of the negative covariances far from center result from finite sample size but not all.

For comparison, we also calculate the normalized Frobenius difference for null models, the unstructured covariance model and a translation invariant V1 model. In the translation invariant model, we remove the spatially localizing exponential in Eq (9) and only fit the spatial frequency parameter, $f$. For the unstructured model, the Frobenius norm difference is 0.9835 while that of the translation invariant model is 0.9727. The V1 covariance model has a much lower difference (0.2993) and is a better fit to the data. We show the covariance matrices and sampled receptive fields from these null models in Fig H to J in S1 Appendix.

Similar spectral properties are evident in the eigenvectors and eigenvalues of $C_{model}$, $C_{finite}$, $C_{data}$, and the analytical forms derived in Eq (10) (Fig 5D and 5E). The covariances are again normalized to have unit trace. Note that the analytical eigenfunctions are shown on a finer grid than the model and data because the analysis was performed in continuous space. The differences between the eigenfunctions and eigenvalues of the analytical and model results are due to discretization. Examining the eigenvectors (Fig 5D), we also see a good match, although there are some rotations and differences in ordering. These 10 eigenvectors explain 68% of the variance in the receptive field data. For reference, the top 80 eigenvectors explain 86% of the variance in the data and all of the variance in the model. The eigenvalue curves of both the models and the analytical forms match that of the data (Fig 5E) reasonably well, although not as well as for the mechanosensors. In S1 Appendix, we repeat this analysis for receptive fields measured with different stimulus sets in the mouse and different experimental dataset from non-human primate V1. Our findings are consistent with the results shown above (Fig K to P in S1 Appendix).

## Advantages of structured random weights for artificial learning tasks

Our hypothesis is that neuronal inductive bias from structured receptive fields allows networks to learn with fewer neurons, training examples, and steps of gradient descent for classification tasks with naturalistic inputs. To examine this hypothesis, we compare the performance of structured receptive fields against classical ones on several classification tasks. We find that, for most artificial learning tasks, structured random networks learn more accurately from smaller network sizes, fewer training examples, and gradient steps.

**Frequency detection.**   CS naturally encode the time-history of strain forces acting on the insect body and sensors inspired by their temporal filtering properties have been shown to accurately classify spatiotemporal data [52]. Inspired by this result, we test sensilla-inspired mechanosensory receptive fields on a timeseries classification task (Fig 6A, top). Each example presented to the network is a 100 ms timeseries sampled at 2 kHz so that $d = 200$, and the goal is to detect whether or not each example contains a sinusoidal signal. The positive examples are sinusoidal signals with $f_1 = 50$ Hz and corrupted by noise so that their SNR = 1.76 (2.46 dB). The negative examples are Gaussian white noise with matched amplitude to the positive examples. Note that this frequency detection task is not linearly separable because of the random phases in positive and negative examples. See S1 Appendix for additional details including the definition of SNR and how cross-validation was used to find the optimal parameters $f_{lo} = 10$ Hz, $f_{hi} = 60$ Hz, and $\gamma = 50$ ms.

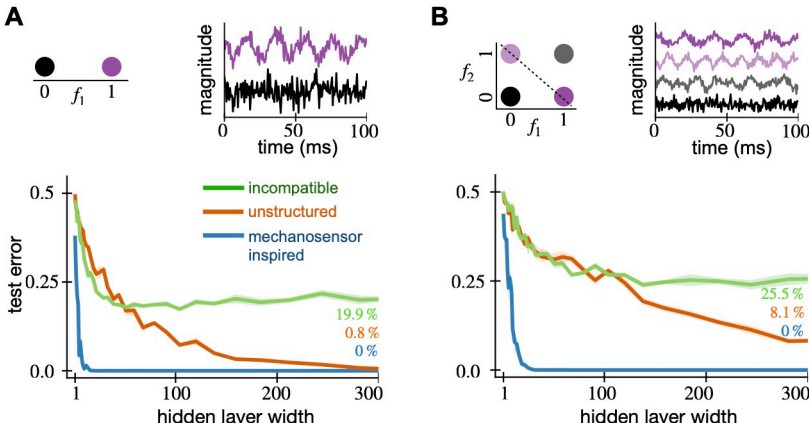

**Fig 6. Random mechanosensory weights enable learning with fewer neurons in time-series classification tasks.** We show the test error of random feature networks with both mechanosensory and classical white-noise weights against the number of neurons in their hidden layer. For every hidden layer width, we generate five random networks and average their test error. In the error curves, the solid lines show the average test error while the shaded regions represent the standard error across five generations of the random network. The top row shows the timeseries tasks that the networks are tested on. (A, top) In the frequency detection task, a $f_1 = 50$ Hz frequency signal (purple) is separated from white noise (black). (B, top) In the frequency XOR task, $f_1 = 50$ Hz (purple) and $f_2 = 80$ Hz (light purple) signals are separated from white noise (black) and mixtures of 50 Hz and 80 Hz (gray). When their covariance parameters are tuned properly, mechanosensor-inspired networks achieve lower error using fewer hidden neurons on both frequency detection (A, bottom) and frequency XOR (B, bottom) tasks. However, the performance of bio-inspired networks suffer if their weights are incompatible with the task.

For the same number of hidden neurons, the structured RFN significantly outperforms a classical RFN. We show test performance using these tuned parameters in Fig 6A. Even in this noisy task, it achieves 0.5% test error using only 25 hidden neurons. Meanwhile, the classical network takes 300 neurons to achieve similar error.

Predictably, the performance suffers when the weights are *incompatible* with the task. We show results when $f_{lo} = 10$ Hz and $f_{hi} = 40$ Hz and the same $\gamma$ (Fig 6A). The incompatible RFN performs better than chance (50% error) but much worse than the classical RFN. It takes 300 neurons just to achieve 16.3% test error. The test error does not decrease below this level even with additional hidden neurons.

**Frequency XOR task.** To challenge the mechanosensor-inspired networks on a more difficult task, we build a frequency Exclusive-OR (XOR) problem (Fig 6B, top). XOR is a binary function which returns true if and only if the both inputs are different, otherwise it returns false. XOR is a classical example of a function that is not linearly separable and thus harder to learn. Our inputs are again 100 ms timeseries sampled at 2 kHz. The inputs either contain a pure frequency of $f_1 = 50$ Hz or $f_2 = 80$ Hz, mixed frequency signals with both $f_1$ and $f_2$, or white noise. In both the pure and mixed frequency cases, we add noise so that the SNR = 1.76. See S1 Appendix for details. The goal of the task is to output true if the input contains either pure tone and false if the input contains mixed frequencies or is white noise.

We tune the GP covariance parameters $f_{lo}$, $f_{hi}$, and $\gamma$ from Eq (8) using cross-validation. The cross validation procedure and algorithmic details are identical to that of the frequency detection task. Using cross validation, we find the optimal parameters to be $f_{lo} = 50$ Hz, $f_{hi} = 90$ Hz, and $\gamma = 40$ ms. For incompatible weights, we take $f_{lo} = 10$ Hz, $f_{hi} = 60$ Hz, and the same $\gamma$.

The structured RFN significantly outperform classical RFN for the same number of hidden neurons. We show network performance using these parameters in Fig 6B. Classification error of 1% can be achieved with 25 hidden neurons. In sharp contrast, the classical RFN requires 300 hidden neurons just to achieve 6% error. With incompatible weights, the network needs

300 neurons to achieve just 15.1% test error and does not improve with larger network sizes. Out of the four input subclasses, it consistently fails to classify pure 80 Hz sinusoidal signals which are outside its passband.

**Image classification.**   We next test the V1-inspired receptive fields on two standard digit classification tasks, MNIST [53] and KMNIST [54]. The MNIST and KMNIST datasets each contain 70,000 images of handwritten digits. In MNIST, these are the Arabic numerals 0–9, whereas KMNIST has 10 Japanese *hiragana* phonetic characters. Both datasets come split into 60,000 training and 10,000 test examples. With 10 classes, there are 6,000 training examples per class. Every example is a 28 × 28 grayscale image with centered characters.

Each hidden weight has its center $c$ chosen uniformly at random from all pixels. This ensures that the network's weights uniformly cover the image space and in fact means that the network can represent any sum of locally-smooth functions (see S1 Appendix). We use a network with 1,000 hidden neurons and tune the GP covariance parameters $s$ and $f$ from Eq (9) using 3-fold cross validation on the MNIST training set. Each parameter ranges from 1 to 20 pixels, and the optimal parameters are found with a grid search. We find the optimal parameters to be $s = 5$ pixels and $f = 2$ pixels. We then refit the optimal model using the entire training set. The parameters from MNIST were used on the KMNIST task without additional tuning.

The V1-inspired achieves much lower average classification error as compared to the classical RFN for the same number of hidden neurons. We show learning performance using these parameters on the MNIST task in Fig 7A. To achieve 6% error on the MNIST task requires 100 neurons versus 1,000 neurons for the classical RFN, and the structured RFN achieves 2.5% error with 1,000 neurons. Qualitatively similar results hold for the KMNIST task (Fig 7B),

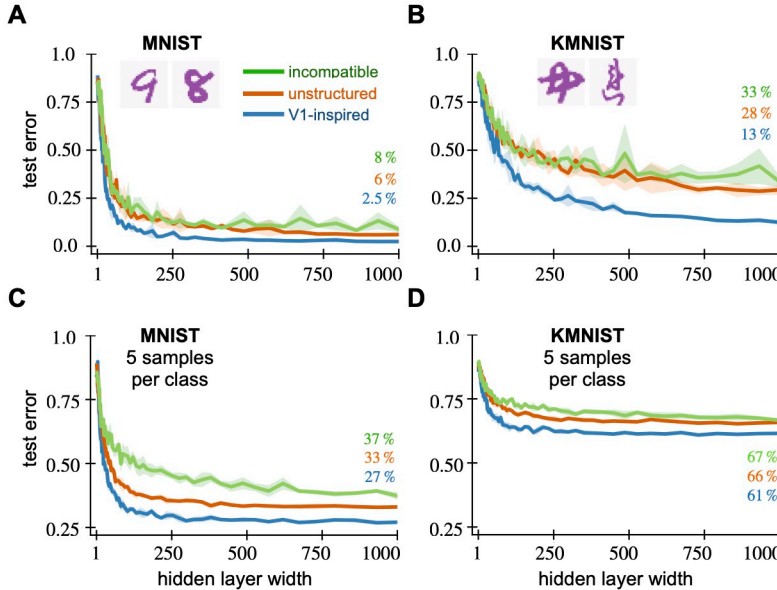

**Fig 7. Random V1 weights enable learning with fewer neurons and fewer examples on digit classification tasks.** We show the average test error of random feature networks with both V1 and classical white-noise weights against the number of neurons in their hidden layer. For every hidden layer width, we generate five random networks and average their test error. The solid lines show the average test error while the shaded regions represent the standard error across five generations of the random network. The top row shows the network's test error on (A) MNIST and (B) KMNIST tasks. When their covariance parameters are tuned properly, V1-inspired networks achieve lower error using fewer hidden neurons on both tasks. The network performance deteriorates when the weights are incompatible to the task. (C) MNIST and (D) KMNIST with 5 samples per class. The V1 network still achieves lower error on these fewshot tasks when the parameters are tuned properly.

although the overall errors are larger, reflecting the harder task. To achieve 28% error on KMNIST requires 100 neurons versus 1,000 neurons for the classical RFN, and the structured RFN achieves 13% error with 1,000 neurons.

Again, network performance suffers when GP covariance parameters do not match the task. This happens if the size parameter $s$ is smaller than the stroke width or spatial scale $f$ doesn't match the stroke variations in the character. Taking the incompatible parameters $s = 0.5$ and $f = 0.5$ (Fig 7A and 7B), the structured weights performs worse than the classical RFN in both tasks. With 1,000 hidden neurons, it achieves the relatively poor test errors of 8% on MNIST (Fig 7A) and 33% on KMNIST (Fig 7B).

**Structured weights improve generalization with limited data.**   Alongside learning with fewer hidden neurons, V1 structured RFNs also learn more accurately from fewer examples. We test few-shot learning using the image classification datasets. The training examples are reduced from 60,000 to 50, or only 5 training examples per class. The test set and GP parameters remain the same.

Structured encodings allow learning with fewer samples than unstructured encodings. We show these few-shot learning results in Fig 7C and 7D. The networks' performance saturate past a few hundred hidden neurons. For MNIST, the lowest error achieved by V1 structured RFN is 27% versus 33% for the classical RFN and 37% using incompatible weights (Fig 7C). The structured network acheives 61% error using structured features on the KMNIST task, as opposed to 66% for the classical RFN and 67% using incompatible weights (Fig 7D).

**Networks train faster when initialized with structured weights.**   Now we study the effect of structured weights as an initialization strategy for fully-trained neural networks where all weights in the network vary. We hypothesized that structured initialization allows networks to learn faster, i.e. that the training loss and test error would decrease faster than with unstructured weights. We have shown that the performance of RFNs improves with biologically inspired weight sampling. However, in RFNs Eq (1) only the readout weights $\boldsymbol{\beta}$ are modified with training, and the hidden weights $W$ are frozen at their initial value.

We compare the biologically-motivated initialization with a classical initialization where the variance is inversely proportional to the number of hidden neurons, $\boldsymbol{w}_{\text{unstruct}} \sim \mathcal{N}\left(0, \frac{2}{d}\boldsymbol{I}\right)$. This initialization is widely known as the "Kaiming He normal" scheme and is thought to stabilize training dynamics by controlling the magnitude of the gradients [55]. The classical approach ensures that $\text{Tr}\left(\frac{2}{d}\boldsymbol{I}\right) = 2$, so for fair comparison we scale our structured weight covariance matrix to have $\text{Tr}(\boldsymbol{C}) = 2$. In our studies with RFNs the trace is equal to $d$, but this weight scale can be absorbed into the readout weights $\boldsymbol{\beta}$ due to the homogeneity of the ReLU.

We again compare structured and unstructured weights on MNIST and KMNIST tasks, common benchmarks for fully-trained networks. The architecture is a single hidden layer feedforward neural network (Fig 1) with 1,000 hidden neurons. The cross-entropy loss over the training sets are minimized using simple gradient descent (GD) for 3,000 epochs. For a fair comparison, the learning rate is optimized for each network separately. We define the area under the training loss curve as a metric for the speed of learning. Then, we perform a grid search in the range of $(1e^{-4}, 1e^0)$ for the learning rate that minimizes this metric, resulting in the parameters 0.23, 0.14, 0.14 for structured, unstructured and incompatible networks respectively. All other parameters are the same as for image classification.

In both the MNIST and KMNIST tasks, the V1-initialized network minimizes the loss function faster than the classically initialized network. For the MNIST task, the V1 network achieves a loss value of 0.05 after 3,000 epochs compared to 0.09 for the other network (Fig 8A). We see qualitatively similar results for the KMNIST task. At the end of training, the V1-inspired network's loss is 0.08, while the classically initialized network only reaches 0.12

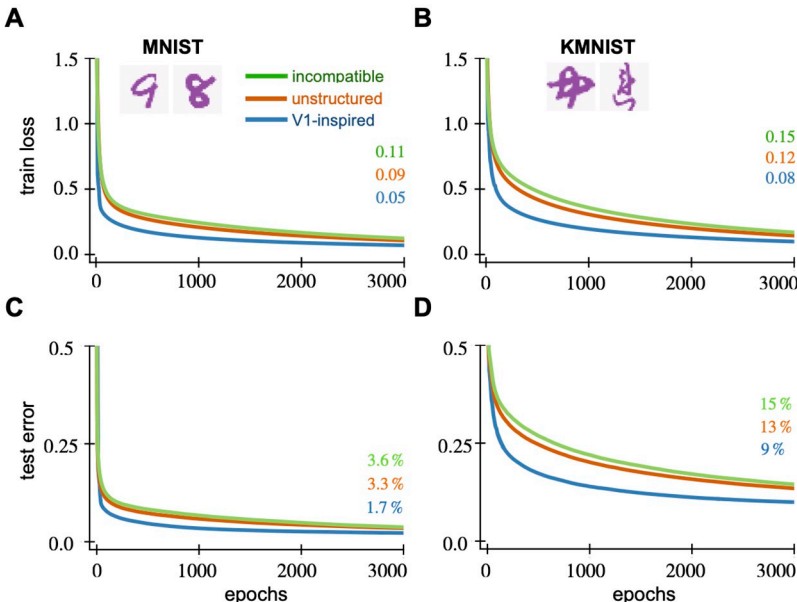

**Fig 8. V1 weight initialization for fully-trained networks enables faster training on digit classification tasks.** We show the average test error and the train loss of fully-trained neural networks against the number of training epochs. The hidden layer of each network contains 1,000 neurons. We generate five random networks and average their performance. The solid lines show the average performance metric across five random networks while the shaded regions represent the standard error. The top row shows the network's training loss on (A) MNIST and (B) KMNIST tasks. The bottom row shows the corresponding test error on (C) MNIST and (D) KMNIST tasks. When their covariance parameters are tuned properly, V1-initialized networks achieve lower training loss and test error under fewer epochs on both MNIST and KMNIST tasks. The network performance is no better than unstructured initialization when the weights are incompatible with the task.

(Fig 8B). We find that the the V1-initialized network performs no better than classical initialization when the covariance parameters do not match the task. With incompatible parameters, the V1-initialized network achieves a loss value of 0.11 on MNIST and 0.15 on KMNIST.

Not only does it minimize the training loss faster, the V1-initialized network also generalizes well and achieves a lower test error at the end of training. For MNIST, it achieves 1.7% test error compared to 3.3% error for the classically initialized network, and 3.6% using incompatible weights (Fig 8C). For KMNIST, we see 9% error compared to 13% error with classical initalization and 15% using incompatible weights (Fig 8D).

We see similar results across diverse hidden layer widths and learning rates (Fig Q to T in S1 Appendix), with the benefits most evident for wider networks and smaller learning rates. Furthermore, the structured weights show similar results when trained for 10,000 epochs (rate 0.1; 1,000 neurons; not shown) and with other optimizers like minibatch Stochastic Gradient Descent (SGD) and ADAM (batch size 256, rate 0.1; 1,000 neurons; not shown). Structured initialization facilitates learning across a wide range of networks.

However, the improvement is not universal: no significant benefit was found by initializing the early convolutional layers of the deep network AlexNet [56] and applying it to the ImageNet dataset [57], as shown in S1 Appendix and Fig U in S1 Appendix. The large amounts of training data and the fact that only a small fraction of the network was initialized with structured weights could explain this null result. Also, in many of these scenarios the incompatible structured weights get to performance on par with the compatible ones by the end of training, when the poor inductive bias is overcome.

**Improving representation with structured random weights.** We have shown how structured receptive field weights can improve the performance of RFNs and fully-trained networks on a number of supervised learning tasks. As long as the receptive fields are compatible with the task itself, then performance gains over unstructured features are possible. If they are incompatible, then the networks performs no better or even worse than using classical unstructured weights.

These results can be understood with our theoretical framework. Structured weights effectively cause the input $x$ to undergo a linear transformation into a new representation $\tilde{x}$ following Theorem 1. In all of our examples, this new representation is bandlimited due to how we design the covariance function. (The V1 weights have all eigenvalues nonzero, but the spectrum decays exponentially, so it acts as a lowpass filter.) By moving to a bandlimited representation, we both filter out noise—high-frequency components—and reduce dimensionality—coordinates in $\tilde{x}$ outside the passband are zero. In general, noise and dimensionality both make learning harder.

It is easiest to understand these effects in the frequency detection task. For simplicity, assume we are using the stationary features of our warm-up to do frequency detection. In this task, all of the signal power is contained in the $f_1 = 50$ Hz frequency, and everything else is due to noise. If the weights are compatible with the task, this means that $w$ is a sum of sines and cosines of frequencies $\omega_k$ in some passband which includes $f_1$. The narrower we make this bandwidth while still retaining the signal, the higher the SNR of $\tilde{x}$ becomes since more noise is filtered out (see S1 Appendix).

## Discussion

In this paper, we describe a random generative model for the receptive fields of sensory neurons. Specifically, we model each receptive field as a random filter sampled from a Gaussian process (GP) with covariance structure matched to the statistics of experimental neural data. We show that two kinds of sensory neurons—insect mechanosensory and simple cells in mammalian V1—have receptive fields that are well-described by GPs. In particular, the generated receptive fields, their second-order statistics, and their principal components match with receptive field data. Theoretically, we show that individual neurons perform a randomized transformation and filtering on the inputs. This connection provides a framework for sensory neurons to compute input transformations like Fourier and wavelet transforms in a biologically plausible way.

Our numerical results using these structured random receptive fields show that they offer better learning performance than unstructured receptive fields on several benchmarks. The structured networks achieve higher test performance with fewer neurons and fewer training examples, unless the frequency content of their receptive fields is incompatible with the task. In networks that are fully trained, initializing with structured weights leads to better network performance (as measured by training loss and generalization) in fewer iterations of gradient descent. Structured random features may be understood theoretically as transforming inputs into an informative basis that retains the important information in the stimulus while filtering away irrelevant signals.

### Modeling other sensory neurons and modalities

The random feature formulation is a natural extension of the traditional linear-nonlinear (LN) neuron model. This approach may be applied to other brain regions where LN models are successful, for instance sensory areas with primarily feedforward connectivity like somatosensory and auditory regions. The neurons in auditory and somatosensory systems are selective to

both spatial and temporal structures in their stimuli [10, 11, 58], and spatial structure emerges in networks trained on artificial tactile tasks [59]. Their receptive fields could be modeled by GPs with spatiotemporal covariance functions [60]; these could be useful for artificial tasks with spatiotemporal stimuli such as movies and multivariate timeseries. Neurons with localized but random temporal responses were found to be compatible with manifold coding in a decision-making task [61]. Our GPs are a complementary approach to traditional sparse coding [62] and efficient coding [63, 64] hypotheses; the connections to these other theories are interesting for future research.

## Receptive fields in development

Our generative model offers new directions to explore the biological basis and computational principles behind receptive fields. Development lays a basic architecture that is conserved from animal to animal [65, 66], yet the details of every neural connection cannot be specified [67], leading to some amount of inevitable randomness at least initially [19]. If receptive fields are random with constrained covariance, it is natural to ask how biology implements this. Unsupervised Hebbian dynamics with local inhibition can allow networks to learn principal components of their input [68, 69]. An interesting future direction is how similar learning rules may give rise to overcomplete, nonorthogonal structure similar to what has been studied here. This may prove more biologically plausible than weights that result from task-driven optimization.

The above assumes that receptive field properties actually lie within synaptic weights. For spatial receptive fields, this assumption is plausible [70], but the temporal properties of receptive fields are more likely a result of neurons' intrinsic dynamics, for which the LN framework is just a model [71–73]. Heterogeneous physiological (e.g. resonator dynamics) and mechanical (position and shape of mechanosensor relative to body structure) properties combine to give the diverse temporal receptive field structures [74]. Development thus leverages different mechanisms to build structure into receptive field properties of sensory neurons.

## Connections to compressive sensing

Random projections have seen extensive use in the field of compressive sensing, where a high-dimensional signal can be found from only a few measurements so long as it has a sparse representation [75–77]. Random compression matrices are known to have optimal properties, however in many cases structured randomness is more realistic. Recent work has shown that structured random projections with local wiring constraints (in one dimension) were compatible with dictionary learning [78], supporting previous empirical results [79]. Our work shows that structured random receptive fields are equivalent to employing a wavelet dictionary and dense Gaussian projection.

## Machine learning and inductive bias

An important open question for both neuroscience and machine learning is why certain networks, characterized by features such as their architecture, weights, and nonlinearities, are better than others for certain problems. One perspective is that a network is good for a problem if it is biased towards approximating functions that are close to the target, known as an *inductive bias*, which depends on an alignment between the features encoded by neurons and the task at hand [32]. Our approach shows that structured receptive fields are equivalent to a linear transformation of the input that can build in such biases. Furthermore, we can describe the nonlinear properties of the network using the kernel, which varies depending on the receptive field structure. If the target function has a small norm in this RKHS, then there is an inductive bias

and it is easier to learn [80, 81]. A small norm in the RKHS means that the target function varies smoothly over the inputs. Smooth functions are easier to learn compared to fast varying ones. In this way, the receptive field structure influences the ease of learning of the target function. We conjecture that receptive fields from neural-inspired distributions affect the RKHS geometry such that the target function's norm is small in that RKHS, compared to the RKHS of random white-noise receptive fields. We leave to future work to verify this conjecture in detail.

Networks endowed with principles of neural computation like batch normalization, pooling of inputs, and residual connections have been found to contain inductive biases for certain learning problems [82, 83]. Learning data-dependent kernels is another way to add in inductive bias [84]. We also saw that initializing fully-trained networks from our generative models improved their speed of convergence and generalization compared to unstructured initialization. This result is consistent with known results that initialization has an effect on generalization [85]. The initialization literature has mostly been focused on avoiding exploding/vanishing gradients [55, 86]. Here, we conjecture that the inductive bias in our structured connectivity places the network closer to a good solution in the loss landscape [67].

The random V1-inspired receptive fields that we model can be seen as similar to what happens in a convolutional neural network (CNN) [87], which have similarities and differences compared to brains [88]. A recent study showed that CNNs with a fixed V1-like convolutional layer are more robust to adversarial perturbations to their inputs [89]. In a similar vein to our work, using randomly sampled Gabor receptive fields in the first layer of a deep network was also shown to improve its performance [90]. The wavelet scattering transform is a multi-layer network where wavelet coefficients are passed through nonlinearities, a model which is similar to deep CNNs [91–93]. Our framework differs as a randomized model and yields wavelets of a single scale, and similar studies of robustness and learning in deep networks with our weights are possible. Adding layers to our model or sampling weights with a variety of spatial frequencies and field sizes would yield random networks that behave similar to the scattering transform, offering an another connection between the brain and CNNs. Directly learning filters in a Hermite wavelet basis led to good perfomance in ANNs with little data [37], and this idea was extended to multiple scales by [94]. Our structured random features can be seen as an RFN version of those ideas with supporting evidence that these principles are used in biology.

## Limitations and future directions

There are several limitations to the random feature approach. We model neuron responses with a scalar firing rates instead of discrete spikes, and we ignore complex neuronal dynamics, neuromodulatory context, and many other details. Like most LN models, the random feature model assumes zero plasticity in the hidden layer neurons. However, associative learning can drive changes in receptive fields of individual neurons in sensory areas like V1 and auditory cortex [95, 96]. Further, our RFN is purely feedforward and cannot account for feedback connections. Recent work suggests that feedforward architecture lacks sufficient computational power to serve as a detailed input-output model for a network of cortical neurons; it might need additional layers with convolutional filters [97]. It can be difficult to interpret the parameters found from fitting receptive field data and connect them to experimental conditions. Also, the GP model of weights only captures covariance (second moments) and neglects higher-order statistics. It remains to be shown how the theory can yield concrete predictions that can be tested *in vivo* experimental conditions.

The random feature receptive field model is a randomized extension of the LN neuron model. The LN model fits a parameterized function to each receptive field [6]. In contrast, the

random feature framework fits a distribution to an entire population of receptive fields and generates realistic receptive fields from that distribution. A natural question is how they compare. If the goal is to capture individual differences between neuronal receptive fields, one should resort to an LN model where each neuron's receptive field is fit to data. The random feature model is not as flexible, but it provides a direct connection to random feature theory, and it is mathematically tractable and generative. This connection to kernel learning opens the door to using techniques which are a mainstay in machine learning theory literature, for instance to estimate generalization error and sample complexity [80], in the context of learning in more biologically realistic networks.

We see several future directions of structured random features in connecting computational neuroscience and machine learning. As already stated, the auditory, somatosensory, and tactile regions are good candidates for further study as well as developmental principles that could give rise to random yet structured receptive field properties. To account for plasticity in the hidden layer, one could also analyze the neural tangent kernel (NTK) associated with structured features [98]. These kernels are often used to analyze ANNs trained with gradient descent when the number of hidden neurons is large and the step size is small [26]. To incorporate lateral and feedback connections, the weights could be sampled from GPs with recurrent covariance functions [99]. Our theory may also help explain why CNNs with fixed V1-like convolutional layer are more robust to adversarial input perturbations [89] as filtering out high frequency corruptions. It seems likely that structured random features will also be more robust. It would be interesting to study intermediate layer weights of fully-trained networks as approximate samples from a GP by studying their covariance structure. Finally, one could try and develop other covariance functions and further optimize these RFNs for most sophisticated learning tasks to see if near high performance—lower error, faster training, etc.—on more difficult tasks is possible.

## Methods

The methods are described throughout the Results section. Further details and additional results are in S1 Appendix.

## Supporting information

**S1 Appendix. Additional theory and supporting data. Fig A**: **Simulation results for the simplified frequency detection task**. On the left, test error versus dataset size for kernel ridge regression using structured kernels (purple lines, by bandwidth) and the unstructured kernel (blue). On the right, test accuracy versus dataset size for SVM classifier readout trained on structured random features (purple lines, by bandwidth) and unstructured random features (blue). In both cases, task structural information improves performance, leading to less error. **Fig B**: **Receptive fields of mechanosensory neurons**. We show (A) biological receptive fields and (B) random samples from the fitted covariance model. **Fig C**: **Covariance matrix of mechanosensory receptive fields and unstructured model**. We compare the covariance matrices generated from the (A) receptive fields of 95 mechanosensory neurons, (B) unstructured GP model and (C) 95 random samples from the model. **Fig D**: **Covariance matrix of mechanosensory receptive fields and the Fourier model** (5). We compare the covariance matrices generated from the (A) receptive fields of 95 mechanosensory neurons, (B) Fourier GP model and (C) 95 random samples from the model. **Fig E**: **Receptive fields from mechanosensory neurons, the unstructured model and the Fourier model** (5). We show the receptive fields from the (A) mechanosensory neurons, (B) unstructured GP model and (C) the Fourier GP model. **Fig F**: **Covariance matrix of V1 receptive fields and our model for white**

**noise stimuli**. We show the full structure of the covariance matrices, which are the $180 \times 180$ pixel region around the centers of these $504 \times 504$ pixel matrices. These matrices are generated from the (A) receptive fields of 8,358 mouse V1 neurons, (B) the GP model Eq (9), and (C) 8,358 random samples from the model. **Fig G: Receptive fields of V1 neurons from white noise stimuli**. We show (A) biological receptive fields and (B) random samples from the fitted covariance model. **Fig H: Covariance matrix of V1 receptive fields and unstructured model for white noise stimuli**. We compare the covariance matrices generated from the (A) receptive fields of 8,358 mice V1 neurons, (B) unstructured GP model and (C) 8,358 random samples from the model. **Fig I: Covariance matrix of V1 receptive fields and translation invariant V1 model for white noise stimuli**. We compare the covariance matrices generated from the (A) receptive fields of 8,358 mice V1 neurons, (B) translation invariant version of the V1 GP model and (C) 8,358 random samples from the model. **Fig J: Receptive fields from V1 neurons, the unstructured model and the translation invariant V1 model for white noise stimuli**. We show the receptive fields from the (A) V1 neurons, (B) unstructured GP model and (C) the translation invariant V1 GP model. **Fig K: Spectral properties of V1 receptive fields and our model for Ringach dataset**. We compare the covariance matrices generated from the (A) receptive fields of 250 macaque V1 neurons, (B) the GP model Eq (9), and (C) 250 random samples from the model. The data is from [100]. (D) The leading 10 eigenvectors of the data and model covariance matrices show similar structure and explain 57% of the variance in the data. Analytical Hermite wavlet eigenfunctions are in the last row. (E) The eigenspectrum of the model matches well with the data. **Fig L: Receptive fields of V1 neurons from the Ringach dataset**. We show (A) biological receptive fields and (B) random samples from the fitted covariance model. **Fig M: Spectral properties of V1 receptive fields and our model for natural image stimuli**. We compare the covariance matrices generated from the (A) receptive fields of 10,782 mice V1 neurons, (B) the GP model Eq (9), and (C) 10,782 random samples from the model. (D) The leading 10 eigenvectors of the data and model covariance matrices show similar structure and explain 39% of the variance in the data. Analytical Hermite wavelet eigenfunctions are in the last row (E) The eigenspectrum of the model compared to the data. **Fig N: Receptive fields of V1 neurons from natural images stimuli**. We show (A) biological receptive fields and (B) random samples from the fitted covariance model. **Fig O: Spectral properties of V1 receptive fields and our model for DHT stimuli**. We compare the covariance matrices generated from the (A) receptive fields of 2,698 mice V1 neurons, (B) the GP model Eq (9), and (C) 2,698 random samples from the model. (D) The leading 10 eigenvectors of the data and model covariance matrices. They explain 29% of the variance in the data. Analytical Hermite wavelet eigenfunctions are in the last row. (E) The eigenspectrum of the model matches well with the data. **Fig P: Receptive fields of V1 neurons from DHT stimuli**. We show (A) biological receptive fields and (B) random samples from the fitted covariance model. **Fig Q: Training loss on MNIST for fully-trained neural networks initialized with V1 weights**. We show the average training loss of fully-trained networks against the number of training epochs across diverse hidden layer widths (50, 100, 400, and 1000) and learning rates ($10^{-3}$, $10^{-2}$, and $10^{-1}$). For every hidden layer width, we generate five random networks and average their performance. The solid lines show the average training loss while the shaded region represents the standard error. When the covariance parameters are tuned properly, V1-initialized networks achieve lower training loss over fewer epochs. The benefits are more significant at larger network widths and lower learning rates. With incompatible weights, V1 initialization leads to similar performance as unstructured initialization. **Fig R: Test error on MNIST for fully-trained neural networks initialized with V1 weights**. We show the average test error of fully-trained networks against the number of training epochs across diverse hidden layer widths (50, 100, 400, and 1000) and learning rates ($10^{-3}$, $10^{-2}$, and $10^{-1}$). For every

hidden layer width, we generate five random networks and average their performance. The solid lines show the average test error while the shaded regions represent the standard error. When the covariance parameters are tuned properly, V1-initialized networks achieve lower test error over fewer epochs. The benefits are more significant at larger network widths and lower learning rates. With incompatible weights, V1 initialization leads to similar performance as unstructured initialization. **Fig S: Training loss on KMNIST for fully-trained neural networks initialized with V1 weights**. We show the average training loss of fully-trained networks against the number of training epochs across diverse hidden layer widths (50, 100, 400, and 1000) and learning rates ($10^{-3}$, $10^{-2}$, and $10^{-1}$). For every hidden layer width, we generate five random networks and average their performance. The solid lines show the average training loss while the shaded regions represent the standard error. When the covariance parameters are tuned properly, V1-initialized networks achieve lower training loss over fewer epochs. The benefits are more significant at larger network widths and lower learning rates. With incompatible weights, V1 initialization leads to similar performance as unstructured initialization. **Fig T: Test error on KMNIST for fully-trained neural networks initialized with V1 weights**. We show the average test error of fully-trained networks against the number of training epochs across diverse hidden layer widths (50, 100, 400, and 1000) and learning rates ($10^{-3}$, $10^{-2}$, and $10^{-1}$). For every hidden layer width, we generate five random networks and average their performance. The solid lines show the average test error while the shaded regions represent the standard error. When the covariance parameters are tuned properly, V1-initialized networks achieve lower test error over fewer epochs. The benefits are more significant at larger network widths and lower learning rates. With incompatible weights, V1 initialization leads to similar performance as unstructured initialization. **Fig U: Initializing AlexNet using structured random features shows little benefit for ImageNet**. Training and testing loss are shown for classical and structured random initializations of convolutional layers in AlexNet. These losses are initially lower for structured features, but by 6 epochs the classical initialization catches up and it eventually reaches a slightly lower loss than the structured initialization. Note that the training losses are higher than testing due to dropout applied in the training phase.
(PDF)

## Acknowledgments

We thank Dario Ringach for providing the macaque V1 data and Brandon Pratt for the hawk-moth mechanosensor data. We are grateful to Ali Weber, Steven Peterson, Owen Levin, and Alice C. Schwarze for useful discussions. We thank Sarah Lindo, Michalis Michaelos, and Carsen Stringer for help with mouse surgeries, calcium imaging, and data processing, respectively.

## Author Contributions

**Conceptualization:** Kameron Decker Harris.

**Data curation:** Marius Pachitariu.

**Formal analysis:** Biraj Pandey, Bingni W. Brunton, Kameron Decker Harris.

**Investigation:** Biraj Pandey, Kameron Decker Harris.

**Resources:** Bingni W. Brunton.

**Software:** Biraj Pandey.

**Supervision:** Bingni W. Brunton, Kameron Decker Harris.

**Validation:** Biraj Pandey.

**Visualization:** Biraj Pandey, Bingni W. Brunton, Kameron Decker Harris.

**Writing – original draft:** Biraj Pandey, Kameron Decker Harris.

**Writing – review & editing:** Biraj Pandey, Marius Pachitariu, Bingni W. Brunton, Kameron Decker Harris.

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
