## [Decision Letter · Decision Letter 0]

2 Mar 2022

Dear Mr. Pandey,

Thank you very much for submitting your manuscript "Structured random receptive fields enable informative sensory encodings" for consideration at PLOS Computational Biology.

As with all papers reviewed by the journal, your manuscript was reviewed by members of the editorial board and by several independent reviewers. In light of the reviews (below this email), we would like to invite the resubmission of a significantly-revised version that takes into account the reviewers' comments.

We cannot make any decision about publication until we have seen the revised manuscript and your response to the reviewers' comments. Your revised manuscript is also likely to be sent to reviewers for further evaluation.

Sincerely,

Xuexin Wei

Associate Editor

PLOS Computational Biology

Thomas Serre

Deputy Editor

PLOS Computational Biology

Reviewer's Responses to Questions

**Comments to the Authors:**

Reviewer #1: The authors make two main contributions. First is modeling receptive fields as independent draws from a Gaussian process. Second is arguing that such structured receptive fields are beneficial for efficient learning downstream. Further, authors prove a theorem that provides an interpretation of the random receptive fields as performing filtering in the eigenbasis of the Gaussian process. This "inductive bias" is argued to be the reason why a downstream predictors performance may be enhanced if the receptive fields are compatible with the learning problem.

While the these contributions are potentially important, neither of the two ideas are sufficiently motivated or developed. Therefore, the paper feels light on content. Below are some suggestions for improvement.

1) Why is there a need for modeling receptive fields as random draws? This is not motivated at all. I found it strange that while the introduction of the paper reviews a wide range of literature (including even technical references to reproducing kernel Hilbert spaces whose relevance is questionable), the main motivation for this work is through a short sentence referring to data in ref. [8]. I went through that paper, and did not see how it motivates the authors' approach.

2) What is the advantage of modeling receptive fields as random draws? While authors provide some evidence that their models capture second order statistics of experimental receptive fields, it is not clear what the advantage of this modeling approach is compared to, say, fitting an LN model to every neuron. The authors should state some performance metrics and make comparisons to other relevant techniques.

3) Authors mention RKHSs a few times, and there is even some theoretical discussion in Appendix A.1. However, it is not clear how this connection is important or relevant. Reading the paragraph below equation 12 (sorry, no line numbers are provided), and discussion section 3.4, it feels like the authors want to claim that their learning efficiency results are related to learning in RKHSs. It would have been nice to see this point fully developed.

4) Instead of MNIST, authors could use a dataset with more naturalistic and complex stimuli.

5) A theoretical study of what makes a receptive field covariance compatible or incompatible with downstream learning should be pursued. This may be related to the RKHS picture that the authors are hinting at.

6) The authors have a great list of possible future directions. One or more of these directions could be pursued for this paper.

Other:

1) Black line in Figure 5D is not visible.

Reviewer #2: In the submitted article, the authors present a novel framework for modeling receptive fields in sensory cortices. By introducing the concept of a structured receptive field drawn from a biased probability distribution, they can model sensory neurons in an interpretable fashion while still incorporating the randomness crucial to biological systems. The article is broadly split into two parts: First, structured receptive fields are used to fit experimental data of biological receptive fields, and then the suitability for machine learning applications is discussed.

These results are interesting and novel, but, as described in the points below, there were some respects in which the analysis and explication of the results could be improved.

Major points

--

1. The results on the matching to biological receptive fields are based on qualitative--not quantitative--comparisons. The authors compare the covariance matrices from experimental data with those fitted by the model only based on how similar they look to each other on a plot. One way in which the results might be strengthened is by including a null hypothesis, like purely random receptive fields, or receptive fields based on Gabor filters for V1.

2. The authors do a nice job of motivating their theory with the theory of random feature networks from the machine learning literature. I would also have liked to see the authors discuss in further detail their ideas in the context of previous computational neuroscience approaches to describing receptive fields, including efficient coding, sparse coding, and related approaches.

3. For establishing that structured weights lead to faster learning in Section 2.3.5, a more convincing way to show this in our view would be to (i) define a metric for speed of learning (e.g. area under the learning curve) (ii) optimize the learning rate separately for each network with respect to this metric, and (iii) compare this metric for the two networks with their separately optimized learning rates. The way that the simulations are currently being done leaves open the possibility that the slower learning in the unstructured case might be due to a suboptimal choice of learning rate.

Minor points

--

4. In section 2.1.1, can the authors explain why the L2 regularization parameter was set to 1? Is it possible that any of the presented results would change if this parameter were set differently?

5. In the first paragraph of Section 2.1.3, the claim is made, without specifying any particular task, that adding an inductive bias improves learning. The authors should be careful about statements like this here and elsewhere in the paper since, as they are obviously aware given that they illustrate this point themselves in the simulations, an inductive bias can only be expected to improve learning if it is well matched to the particular computational task, and it can actually worsen learning if this is not the case.

6. In Section 2.1.3, the authors state "the GP covariance function reflects the statistical regularities within the sensory inputs to the network". Can the authors provide a citation for this statement in the cases considered?

7. In the second-to-last paragraph of section 2.2.1, the bandwidth should be f_hi - f_lo.

8. In equation (7) of section 2.2.1, why are the stochastic coefficients for cos(\\omega_k t) and sin(\\omega_k t) not independent?

9. At the end of section 2.2.1, can the authors more precisely explain the statement 'The smoothness is also controlled by the overall magnitude of the nonzero eigenvalues'?

10. In describing the computation of C_data in Section 2.2.2, an explanation of what is meant by 'centered receptive fields' could be helpful.

11. In Section 2.2.3, the unit pixels^2 seems incorrect, since a squared pixel is what is generally meant by a pixel.

12. More explanation of what mechanism leads to the prominent dark patches in the covariance matrices shown in Figure 5 would be appreciated.

13. For the result in Figures 5E (and, to an extent, in Figure 3A), what do the authors mean by 'remarkably well'? This might be addressed by the response to major point 1. For Figure 5E, can the authors comment on the faster decay in the spectrum of the model? Could this be due to noise or some effect related to the calcium imaging?

14. In the text describing Figure 5A, it is stated that C_data is shown zoomed in, but nothing about this is mentioned in the caption, and no mention of the scale is given, so that one has to look at the supplemental figure to get some (still imprecise) idea of what's going on. Given that the dark patches are still clearly visible in the full covariance plotted in the supplemental figure, I would recommend either showing the full version instead in Figure 5 or describing the zoomed-in version more clearly.

15. In the plots for the machine learning applications, why did the authors add the green curves? The fact that they show higher test errors than the blue ones is tautological, since the parameters for the latter were optimized precisely to minimize this test error. There may be some value in these curves if the point is illustrate that a poorly chosen inductive bias is worse than no inductive bias at all, but this doesn't seem to be borne out by all pairs of green vs. orange curves in the figures, and this doesn't seem to be the point that is made in the text.

16. In the last paragraph of appendix A.1, the authors claim that 'functions with small H-norm are easier to learn than those with larger norm'. Can this statement be made more precise, perhaps for a specific regression algorithm?

Reviewer #3: In this paper, authors considered receptive fields as random samples from parametrized distributions (gaussian process), in particular from 2 following modalities: insect mechanosensors and mammalian visual cortex neurons. Then they demonstrated that these 'random feature' neurons (RFN) remove high frequency noise, and boost signals. Finally, they also show that these RFNs enable efficient learning, both from the number of neurons and training time perspective.

This topic is of general interest to the visual neuroscience and the growing subfield within computational neuroscience intersecting with artificial neural networks. I have some comments and questions about the motivations and implications for the work, attached below.

1. The theoretical analysis on the equivalence between structured weight vector transformation and filtering into Gaussian Processing eigenbasis and doing a random projection onto a spherical random gaussian is interesting. Aside from its connection to the kernel theory of learning, what are the broader implications of this finding, either in sensory neuroscience or deep learning fields? It would be helpful to get more discussion around this topic in a revised version.

2. What is the motivation for using "Gaussian Process" type generative model here, from mathematical and/or neuro- perspectives? Is it possible that sampling random filters from Gaussian Process is not only mathematically tractable, but also more biologically plausible (compared to, say, learning filters from the task-driven optimization)? It would be helpful to get more discussion around this in a revised version.

3. Is it surprising that filter samples from GPs with covariance from the real data match various properties of real neural populations? It might be helpful to have control models which fail to capture such biological fidelity (for example in Fig 3,4) in a revised version.

Minor comment: the name "structured random receptive field", while properly defined in the paper, is a bit vague — initially I thought that "random" here implies stochastic, as in Dapello et al.

Overall, I think the paper is interesting, and I recommend acceptance with revisions.

**Have the authors made all data and (if applicable) computational code underlying the findings in their manuscript fully available?**

Reviewer #1: None

Reviewer #2: Yes

Reviewer #3: None

PLOS authors have the option to publish the peer review history of their article (what does this mean?). If published, this will include your full peer review and any attached files.

Reviewer #1: No

Reviewer #2: No

Reviewer #3: No
---

## [Decision Letter · Decision Letter 1]

11 Aug 2022

Dear Dr. Harris,

We are pleased to inform you that your manuscript 'Structured random receptive fields enable informative sensory encodings' has been provisionally accepted for publication in PLOS Computational Biology.

Best regards,

Xuexin Wei

Associate Editor

PLOS Computational Biology

Thomas Serre

Deputy Editor

PLOS Computational Biology

Reviewer's Responses to Questions

**Comments to the Authors:**

Reviewer #1: I thank the authors for their response. I appreciate the effort they put in this revision. Although they argued many of what I asked to be beyond the scope of their paper (I disagree, but I agree that they are not necessary), they did address my concerns. Clarification and new additions to the paper, including Appendix A2, are adequate. I don't have any further comments or suggestions. I recommend acceptance.

After reading the paper again and the rebuttal, I think I was too negative for recommending rejection in my first review. I apologize for that. I think a more fair assessment would have been "major revision".

Reviewer #2: The authors have done an excellent job of responding to my earlier comments. I am pleased to recommend that the manuscript be accepted for publication. I especially appreciated the new appendix section, which greatly strengthens the mathematical foundations of this work.

Two very minor comments follow below on things I noticed as I was reading the manuscript again, which might help to further improve the paper before final publication.

1. A bar plot as part of Fig 2 illustrating the results for the three models described at line 281, perhaps also including a sketch of what the receptive fields for the two null models look like, would be a helpful way to visually summarize the result. A similar visualization could be helpful in Fig 5.

2. Perhaps I missed it, but, in Figures 6 and 7, how exactly are the percentages reported in each panel calculated? Comparing the orange curves with the green curves, it doesn't appear to me that these numbers correspond to the curves they appear next to in any straightforward way that I could discern.

**Have the authors made all data and (if applicable) computational code underlying the findings in their manuscript fully available?**

Reviewer #1: None

Reviewer #2: Yes

PLOS authors have the option to publish the peer review history of their article (what does this mean?). If published, this will include your full peer review and any attached files.

Reviewer #1: No

Reviewer #2: No

---

## [Editor Report · Acceptance letter]

3 Oct 2022

PCOMPBIOL-D-21-01957R1 

Structured random receptive fields enable informative sensory encodings

Dear Dr Harris,

I am pleased to inform you that your manuscript has been formally accepted for publication in PLOS Computational Biology. Your manuscript is now with our production department and you will be notified of the publication date in due course.

With kind regards,

Anita Estes
